# Acid–Base Status and Cerebral Oxygenation in Neonates: A Systematic Qualitative Review of the Literature

**DOI:** 10.3390/children12111549

**Published:** 2025-11-16

**Authors:** Christian Mattersberger, Bernhard Schwaberger, Nariae Baik-Schneditz, Gerhard Pichler

**Affiliations:** 1Division of Neonatology, Department of Paediatrics, Medical University of Graz, Auenbruggerplatz 34/2, 8036 Graz, Austria; christian.adam.mattersberger@gmail.com (C.M.); bernhard.schwaberger@medunigraz.at (B.S.); nariae.baik@medunigraz.at (N.B.-S.); 2Research Unit for Neonatal Micro- and Macrocirculation, Department of Paediatrics, Medical University of Graz, Auenbruggerplatz 34/2, 8036 Graz, Austria

**Keywords:** acid–base status, base deficit, base excess, bicarbonate, blood gas analysis, cerebral oxygenation, near-infrared-spectroscopy neonates, pH

## Abstract

**Highlights:**

**What are the main findings?**

**What is the implication of the main finding?**

**Abstract:**

Introduction: Blood gas analysis is utilized to assess parameters of oxygenation and ventilation, including acid–base status [pH value, base excess (BE) or base deficit (BD), and bicarbonate (HCO_3_)], to evaluate systemic metabolism status. Acid–base imbalances can have complex effects on the organism, potentially impacting oxygen delivery to tissue. Cerebral oximetry is a non-invasive monitoring technique using near-infrared spectroscopy (NIRS) for the continuous measurement of cerebral tissue oxygenation. The relationship between the acid–base status and cerebral tissue oxygenation in neonates remains unclear. This systematic qualitative review aims to analyze current knowledge of the potential correlations between different acid–base status parameters and cerebral tissue oxygenation measured via NIRS in neonates. Methods: A systematic search of PubMed and Ovid Embase was performed, focusing on cerebral oxygenation, neonates, and acid–base status. Risk of bias was assessed using the ‘‘Risk of Bias for Non-randomized Studies of Exposures’’ (ROBINS-E) instrument. Results: Fifty studies that measured parameters of the acid–base status and cerebral tissue oxygenation in the neonatal period were identified. Seven studies demonstrated a correlation between pH and cerebral tissue oxygenation, while eleven studies found no such correlation. Five studies demonstrated a correlation between the BE/BD and cerebral tissue oxygenation, while six studies found no such correlation. Three studies demonstrated a correlation between HCO_3_ and cerebral tissue oxygenation, while five studies found no such correlation. Discussion: Associations between acid–base status parameters and cerebral tissue oxygenation remain controversial. However, studies with the lowest risk of bias mainly demonstrated no significant correlation between any of the acid–base status parameters and cerebral tissue oxygenation.

## 1. Introduction

Cerebral injuries, such as intraventricular haemorrhage (IVH) or periventricular leukomalacia (PVL), remain persistent challenges in neonatology [1,2,3,4]. The neonatal cerebral autoregulation mechanism plays a crucial role in maintaining stable cerebral perfusion and oxygenation. Impairment of these mechanisms can result in hyperoxic or hypoxic conditions, potentially resulting in irreversible complications, such as IVH or PVL [2,3,5,6,7,8]. IVH and PVL are particularly common in extremely preterm infants, and they are associated with poor neurodevelopmental outcome or death [1,9,10,11]. Understanding the physiology and pathophysiology of neonatal cerebral autoregulation and its influencing factors is essential for the prevention of these irreversible severe complications. A major challenge remains—the timely recognition of impaired neonatal cerebral autoregulation and the subsequent imbalance in cerebral oxygen supply and consumption before cerebral complications occur. Unfortunately, current routine monitoring during the neonatal period, including pulse oximetry, electrocardiogram, and blood pressure measurement, does not assess cerebral oxygen delivery or cerebral oxygen consumption, thereby neglecting potentially critical cerebral oxygenation information [12,13,14,15].

Cerebral oximetry is a continuous, non-invasive, real-time method using near-infrared spectroscopy (NIRS) to detect cerebral tissue oxygenation [cerebral regional oxygen saturation (crSO_2_), tissue oxygenation index (TOI), and fractional tissue oxygen extraction (FTOE)]. Cerebral tissue oxygenation depends on cerebral oxygen delivery, cerebral oxygen consumption, and the arterial–venous volume ratio [16,17]. Therefore, using NIRS might be a promising tool for detecting impairments in cerebral perfusion and oxygenation even when routine monitoring parameters still remain within normal ranges [15,18,19]. Research has identified various variables, including cardiovascular and respiratory parameters, that may influence cerebral tissue oxygenation in neonates [17,20,21,22].

Blood gas analysis is a quick and non-invasive point-of-care method for assessing the acid–base status, identifying insufficient systemic oxygenation, and guiding counter-regulation in hypoxic conditions. Acid–base status parameters can serve as outcome predictors and indicators for interventions in neonates [23,24,25]. The pH level may affect neonatal vascular tone, subsequently influencing cerebral oxygen delivery and thus cerebral tissue oxygenation [26]. Furthermore, acidosis is associated with an increased risk for cerebral complications, such as hypoxic ischemic encephalopathy (HIE), IVH, or seizures [27,28,29]. Parameters such as base excess (BE) or its reversal, the base deficit (BD), and bicarbonate (HCO_3_) indicate the counter-regulation of an impaired acid–base status, and deviations are important predictors of neonatal morbidity, such as HIE, seizures, or respiratory complications [27,28,30,31]. Additionally, the administration of HCO_3_ in neonates with metabolic acidosis is a potential treatment option, although its benefits and harms remain controversially debated [29,32]. There is increasing research interest concerning the impact of acid–base status parameters on cerebral tissue oxygenation in neonates [33,34,35]. However, the effects of acid–base status parameters on the autoregulation mechanism and subsequently on cerebral oxygenation in the neonatal brain are unclear. The aim of this review is to provide an overview of the current literature regarding a potential association between parameters of the acid–base status and cerebral oxygenation in neonates during the neonatal period.

## 2. Materials and Methods

### 2.1. Search Strategy and Selection Criteria

Studies were identified using the stepwise approach outlined in the Preferred Reporting Items for Systematic Reviews and Meta-Analyses (PRISMA 2020) statement [36]. To ensure transparency and reproducibility, we pre-registered our study protocol in the International Prospective Register of Systematic Reviews database (PROSPERO, Registration ID: CRD420250655009).

### 2.2. Eligibility Criteria

Studies were included if they reported cerebral tissue oxygenation measurements with NIRS (crSO_2_ or FTOE), along with acid–base status parameters (pH, BE, BD, and HCO_3_), in neonates during the neonatal period.

### 2.3. Search Strategy

A systematic search was conducted on PubMed NCBI and Ovid Embase to identify studies published in the English language between July 1977—the first description of NIRS application in neonates—and May 2025. The search keywords included the following: near-infrared spectroscopy, fractional tissue oxygen extraction, regional cerebral tissue oxygen saturation, oxygenation, term neonates, preterm neonates, newborns, baby, caesarean delivery, vaginal delivery, transition, after birth, neonatal transition, metabolism, pH value, bicarbonate, base-excess, base-deficit, acidosis, alkalosis, acid-base balance, and acid-base imbalance.

### 2.4. Inclusion and Exclusion Criteria—Population

To be eligible, studies had to investigate human neonates with a postnatal age of less than 28 days. Studies that included both neonates with a postnatal age of less than 28 days and older infants or children were also included. Animal studies and studies without an available abstract were excluded.

### 2.5. Inclusion and Exclusion Criteria—Measurements (Exposure)

Studies using any NIRS device for cerebral oximetry were included if any additional measurements of neonatal capillary, venous, or arterial blood pH, BE/BD, or HCO_3_ values were reported.

### 2.6. Inclusion and Exclusion Criteria—Types of Publication

We included all studies published in the English language, excluding non-original articles, such as comments, book chapters, editorials, reviews, and methodological papers. Duplicates and publications in non-English languages were also excluded.

### 2.7. Study Selection

The articles identified in the literature review were independently evaluated by two authors (C.M. and G.P.) based on the titles and the abstracts. Full texts were then retrieved according to the eligibility criteria. In cases of uncertainty regarding inclusion based on the abstract, the full text was also assessed. Any disagreements were resolved through discussion and consensus between the two authors. Data were qualitatively analysed, including the study design, study population (preterm/term neonates), number of neonates, NIRS device used, NIRS measurement time point and duration, acid–base status parameter measurement time point, values of NIRS parameters and acid–base status parameters, the presence or absence of association, and the direction of correlation if an association was detected.

### 2.8. Risk of Bias in Individual Studies

During the planning process of this review, we primarily expected non-randomized observational studies. The bias assessment of non-randomized observational studies through standard bias assessment tools, such as the Newcastle–Ottawa Scale, resulted in a lack of feasibility. Hence, the Risk of Bias in Non-randomized Studies of Exposures (ROBINS-E) tool was used for studies presenting data on associations between parameters of the acid–base status and cerebral oxygenation [37]. The assessment process was conducted using the standardized ROBINS-E Excel implementation. ROBINS-E is a tool for assessing the risk of bias in non-randomized studies and includes seven items: (I) risk of bias due to confounding; (II) risk of bias arising from measurement of the exposure; (III) risk of bias in the selection of participants for the study; (IV) risk of bias due to post-exposure interventions; (V) risk of bias due to missing data; (V.) risk of bias arising from measurements of the outcome; and (VII) risk of bias in the selection of the reported result [36]. Each bias item was rated as low, some concerns, high risk, or very high risk of bias. Confounding factors were defined as follows: i. impaired cerebral autoregulation mechanism; ii. disorders of the cardiovascular system; iii. disorders of the respiratory system; iv. disorders resulting in increased oxygen consumption; and v. congenital malformation. Finally, a complete risk-of-bias rating was assigned to each study for the given answer on the observed presence or absence of association. Studies were categorized as low/some concerns (=low) risk of bias and high/very high (=high) risk of bias to address potential confounding.

## 3. Results

From the preliminary search, 4132 abstracts were identified and assessed for eligibility. Following the full-text review, 50 studies met the inclusion criteria for this systematic review [26,34,38,39,40,41,42,43,44,45,46,47,48,49,50,51,52,53,54,55,56,57,58,59,60,61,62,63,64,65,66,67,68,69,70,71,72,73,74,75,76,77,78,79,80,81,82,83,84,85] (Figure 1).

In total, 21 studies [38,39,40,41,44,45,48,49,50,52,53,57,60,62,66,67,78,79,80,82,83] analyzed two parameters, and 9 studies [34,46,54,63,68,69,71,72,85] reported on all three parameters of the acid–base status in combination with NIRS measurements during the neonatal period. Table 1, Table 2 and Table 3 provide an overview of the basic data of all included studies.

### 3.1. pH Value and Cerebral Tissue Oxygenation

A total of 50 studies reported pH and cerebral NIRS measurements [26,34,38,39,40,41,42,43,44,45,46,47,48,49,50,51,52,53,54,55,56,57,58,59,60,61,62,63,64,65,66,67,68,69,70,71,72,73,74,75,76,77,78,79,80,81,82,83,84,85] (Table 1).

Seven studies reported an association between the pH and cerebral tissue oxygenation (crSO_2_ and FTOE). Of these, four studies [34,38,72,83] found a positive correlation, while three studies [26,69,79] found a negative correlation. Eleven studies demonstrated no association [43,44,46,57,63,64,68,74,75,80,85], while thirty-one studies did not find an association [39,40,41,42,45,47,48,49,50,51,52,53,54,55,56,58,59,60,61,62,65,66,67,70,71,73,76,77,78,81,82].

### 3.2. Base Excess (BE) or Base Deficit (BD) and Cerebral Tissue Oxygenation

A total of 26 studies were identified that reported on BE or BD in combination with cerebral NIRS measurements [34,38,40,41,44,45,46,48,49,50,52,53,54,57,60,62,63,66,68,69,71,72,78,79,83,85] (Table 2).

Five studies found an association between BE and cerebral tissue oxygenation, with four studies [38,69,72,79] reporting a negative correlation and one study [34] reporting a positive correlation. Six studies found no associations [44,46,63,68,83,85], while fifteen studies did not find an association [40,41,45,48,49,50,52,53,54,57,60,62,66,71,78].

### 3.3. Bicarbonate (HCO_3_) and Cerebral Tissue Oxygenation

A total of 13 studies were identified that reported on HCO_3_ measurements in combination with cerebral NIRS measurements [34,39,46,54,63,67,68,69,71,72,80,82,85] (Table 3).

Three studies found an association between HCO_3_ and cerebral tissue oxygenation. Of these, one study [69] reported a negative correlation, while one study [72] reported a positive correlation. Additionally, one study reported a positive correlation between HCO_3_ and fractional tissue oxygen extraction (FTOE) only in term neonates but not in preterm neonates [34]. Five studies demonstrated no associations [46,63,68,80,85], while five studies did not find an association [39,54,67,71,82].

### 3.4. Quality Assessment and Risk of Bias Assessment

Eighteen of these studies provided data on the potential association between acid–base status parameters and cerebral tissue oxygenation [26,34,38,43,44,46,57,63,64,68,69,72,74,75,79,80,83,85]. Quality assessment and risk of bias were assessed for all included studies. Overall, study quality varied (Table 4).

No studies were categorized as low risk. Five studies (27.7%) were categorized as having some concerns, seven studies (38.8%) as high risk, and five studies (33.3%) as very high risk according to the ROBINS-E tool. Table 4 provides an overview of the risk-of-bias assessment. Most included studies were prospective observational studies (66%), followed by retrospective analyses (16%), randomized controlled trials (10%), cross-sectional studies (2%), and case–control studies (4%). An overview of the ROBINS-E assessment and the correlation between the acid–base status parameters and cerebral tissue oxygenation is presented in Table 5.

## 4. Discussion

This systematic qualitative review demonstrates that associations between acid–base status parameters and cerebral tissue oxygenation in neonates are controversial. These conflicting results may arise from different factors, including the diverse populations studied, variations in measurement timing, and differing clinical contexts.

### 4.1. pH Value and Cerebral Tissue Oxygenation

Low pH values may indicate inadequate oxygen supply and/or impaired gas exchange, potentially resulting in irreversible cerebral injury or death [27,28,29]. Studies have reported both negative and positive correlations between pH and cerebral tissue oxygenation. A negative correlation has been observed in neonates, infants, and young children undergoing pediatric heart surgery [26], in those with infantile hypertrophic pyloric stenosis in the perioperative setting [69], and in healthy full-term neonates during the first 10 min of extrauterine life [79]. Conversely, positive correlations have been reported in fetuses shortly before delivery [38]; in preterm and term neonates during the first 15 min after birth [34]; in extremely preterm neonates on the first day after birth [72]; in term-born asphyxiated neonates immediately before the initiation of therapeutic hypothermia [83]; and in preterm neonates with and term neonates without respiratory support during the first 15 min after birth [85]. These discrepancies may be attributed to variations in sample size, timing, and interval between measurements of acid–base status parameters and cerebral tissue oxygenation, and the clinical setting (e.g., HCO_3_ administration, surgical procedures).

### 4.2. Base Excess (BE) or Base Deficit (BD) and Cerebral Tissue Oxygenation

The BE or BD, in combination with the pH, is used alongside clinical and neurological parameters to evaluate neonatal well-being and assess the likelihood and severity of perinatal asphyxia. Furthermore, BE or BD may also assist in guiding therapeutic decisions, such as the initiation of hypothermia therapy for neonates experiencing hypoxic ischemic events [83,86].

Some studies have shown a negative correlation between BD and cerebral tissue oxygenation in fetuses shortly before delivery [38] and in extremely preterm neonates on the first day after birth [72]. Against these, some studies have shown a negative correlation between BE and cerebral tissue oxygenation in neonates and infants suffering from infantile hypertrophic pyloric stenosis in the perioperative setting [69] and in healthy full-term neonates during the first 10 min of extra-uterine life [79]. Finally, one study reported a positive correlation between BE and cerebral tissue oxygenation in preterm and term neonates during the first 15 min after birth [34]. These differences may be due to variations in the number of included neonates, the timing and interval between measurements of acid–base status parameters and cerebral tissue oxygenation, and the clinical setting (e.g., HCO_3_ administration, surgical procedures).

### 4.3. Bicarbonate (HCO_3_) and Cerebral Tissue Oxygenation

HCO_3_ is a critical predictor of neonatal morbidity, and it is used to correct metabolic acidosis, thereby improving hemodynamic parameters. The benefits and harms of HCO_3_ administration in neonates are debated in the literature [34,46,63,68,69,72,80]. Three studies analyzed the effect of HCO_3_ administration on HCO_3_ levels and cerebral tissue oxygenation [34,69,72], whereby two studies [34,72] found a positive association, while one study [69] reported a negative correlation. These differences may be due to variations in the number of included neonates, the gestational age of the neonates, the timing and interval between measurements of acid–base status parameters and cerebral tissue oxygenation, and the clinical setting (e.g., HCO_3_ administration, surgical procedures).

### 4.4. Studies Involving Neonates with Cardiovascular System Impairments

Four studies focused on neonates with congenital heart disease [26,57,64,80]. Except for the study by Amigoni et al. [26], no association was found between acid–base status parameters and cerebral tissue oxygenation in these neonates [57,64,80]. As discussed in the Introduction, the cardiovascular system significantly influences cerebral tissue oxygenation in neonates. Impairments in the cardiovascular system, as observed in neonates with congenital heart diseases, may act as confounders, resulting in potential bias and the absence of observed associations between acid–base status parameters and cerebral tissue oxygenation. This confounding effect might also depend on the severity of congenital malformation, which varied significantly among the included studies. Additionally, two studies [26,57] measured acid–base status parameters and cerebral tissue oxygenation during surgery involving a cardiopulmonary bypass procedure.

### 4.5. Studies During the Transition from Intra- to Extrauterine Life

The immediate postnatal period is characterized by unique physiological conditions. Studies examining neonates during and immediately after the transition from intra- to extrauterine life have reported both positive and negative correlations between acid–base status parameters and cerebral tissue oxygenation [34,38,79]. Notably, in the study by Aldricht et al. [38], cerebral tissue oxygenation was measured during childbirth prior to the clamping of the umbilical cord. Both Aldricht et al. [38] and Mattersberger et al. [34] found positive correlations between acid–base status parameters and cerebral tissue oxygenation in neonates during delivery and immediately after birth, respectively. In contrast, Leroy et al. [79] found a negative correlation in healthy full-term singleton neonates during uncomplicated transition. Mattersberger et al. [34] observed differences in correlations between preterm and term neonates. These differences suggest that gestational age-related differences may, at least in part, explain the variability in correlations observed across studies investigating the immediate transition after birth.

### 4.6. Studies on Term and Preterm Neonates

The physiological differences between preterm and term neonates may impact the relationship between acid–base status parameters and cerebral tissue oxygenation. As already mentioned, Mattersberger et al. [34] found associations between capillary measured pH/BE levels and brain oxygenation in preterm neonates, whereas associations between HCO_3_ and cerebral tissue oxygenation were observed in term neonates. Against them, Dusleag et al. [85] found no associations between the pH/BE/HCO_3_—measured with respect to the umbilical cord blood—and cerebral tissue oxygenation in preterm neonates with and term neonates without respiratory support during the first 15 min after birth [85]. Unfortunately, no other study compared preterm and full-term neonates. However, some studies included preterm neonates [43,46,63,68,72,74], while others included full-term neonates [75,79,83], yielding controversial results (Table 1, Table 2 and Table 3) that suggest that gestational age has an impact on the influence of acid–base status on cerebral tissue oxygenation.

### 4.7. Proposed Explanatory Model

#### 4.7.1. pH and Cerebral Oxygenation

Acidosis can reduce the contractility of cardiomyocytes and diminish cardiac responsiveness to catecholamines, which may both lower cardiac output and cerebral oxygen delivery [87]. However, in hemodynamically stable preterm infants, myocardial function remains largely unaffected during the early transitional period, even with markedly low pH levels, and there appears to be no clear relationship between pH and cardiac output in the first three days after birth [88]. The vascular response to acidosis is developmentally regulated. From day 4 to 14 after the transition, systemic vascular resistance decreases and left ventricular output increases, while no association between pH and vascular tone is seen during the first three days [88]. These observations may demonstrate that the effect of pH on vascular tone—and consequently on regional blood flow—is dependent on postnatal age and therefore differs according to postnatal age. In addition, acidosis causes cerebral vasodilation, resulting in increased cerebral blood flow and oxygenation [26]. These effects could explain an increase in cerebral blood supply and, consequently, cerebral oxygenation.

#### 4.7.2. Base Excess (BE) or Base Deficit (BD) and Cerebral Tissue Oxygenation

A possible explanation for the association between BE or BD and cerebral oxygenation is that BE or BD is calculated from the pH value. Furthermore, BE is an indicator of shock, resuscitation efficacy, and volume deficit, and a decreased BE likely indicates centralization, with resulting hemodynamic effects on cerebral oxygen delivery [89,90].

#### 4.7.3. Bicarbonate (HCO_3_) and Cerebral Tissue Oxygenation

A possible explanation for the correlation between HCO_3_ and cerebral oxygenation may be the strong link between pH and HCO_3_ levels. However, a study in healthy males demonstrated that bicarbonate concentration affects cerebral blood flow (CBF) independently of pH and CO_2_ levels [91].

### 4.8. Risk-of-Bias Assessment of Studies Describing Correlations Between Parameters of Acid–Base Status and Cerebral Oxygenation

Only one study [79], categorized as “low risk of bias” or “some concerns”, demonstrated a correlation of pH level and BE with cerebral tissue oxygenation in neonates. Six studies [26,34,38,69,72,83], categorized as “high risk” or “very high risk of bias”, also described such an association (Table 5). Using the ROBINS-E tool for risk of bias assessment, the main factors for increased risk of bias were a lack of consideration of confounders, selection bias, post-exposure intervention bias, and missing data (Table 4). In some studies, acid–base status parameters were secondary outcome parameters [57,64,68,75,83,85]. These methodological weaknesses may contribute to the incongruent results regarding potential associations.

## 5. Conclusions

This review reveals controversial associations between acid–base status parameters and cerebral tissue oxygenation in neonates. The variability in findings can be attributed to heterogeneity in study populations, especially concerning gestational age, timing of measurements, and clinical contexts. However, the studies with the lowest risk of bias mostly demonstrated the absence of association between the acid–base status parameters and cerebral tissue oxygenation, suggesting the need for more consistent methodologies and further research in this area.

## Figures and Tables

**Figure 1 children-12-01549-f001:**
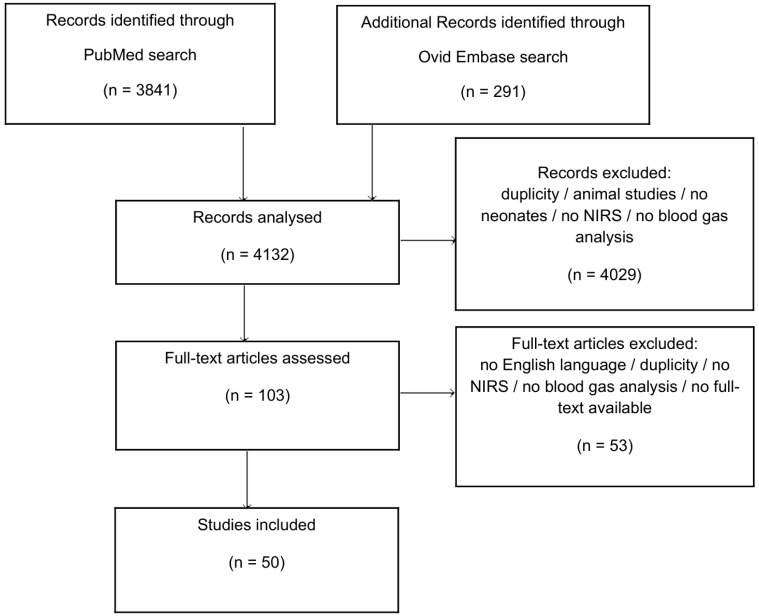
PRISMA flow chart; NIRS = near-infrared spectroscopy.

**Table 1 children-12-01549-t001:** pH and cerebral oxygenation in the neonatal period.

First Author, Years	Study Design	Neonates	n	Device	NIRS Measurement, Time Point	Blood Sample, Time Point	NIRS Measurement, Duration	TOI or crSO_2_ or FTOE	pH, Mean Value	Association, Correlation
Aldrich C.J., 1994 [38]	Prospective observational study	Fetus at delivery	33	n.r.	During the delivery	Immediately after birth	10 min period 30 min before birth	n.r	n.r.	Yes Positive
Naulaers G., 2002 [39]	Observational study	Preterm neonates	15	NIRO 300	Day 1–3 after birth	Before and after NIRS measurements	30 min	Day 1: 57% Day 2: 66.1% Day 3: 76.1%	n.r.	n.r.
Ramamoorthy C., 2002 [40]	Randomized crossover trial	Preterm and term neonates	15	NIM	Day 3 after birth	At the end of base, 17% fractional inspired O_2_, second base, and 3% fractional inspired CO_2_	10–20 min period at base, during 17% fractional inspired O_2_, at second base, and during 3% fractional inspired CO_2_	At base: 53%; 17% fractional inspired O_2_: 53%. At second base: 56%; 3% fractional inspired CO_2_: 68%	At base: 7.43; during 17% fractional inspired O_2_: 7.46; at second base: 7.45; during 3% fractional inspired CO_2_: 7.34	n.r.
Andropoulos D.B., 2003 [41]	Prospective observational study	n.r.	34	INVOS 5100	Day 13 (2–128) after birth	Every 10 to 20 min during bypass	At baseline full cardiopulmonary bypass flow, during low-flow cerebral perfusion, after repair full flow	At baseline: 87%; during low-flow cerebral perfusion: 88%; after full-flow repair: 86%	At baseline: 7.49; during low-flow cerebral perfusion: 7.52; after full-flow repair: 7.44	n.r.
Naulaers G., 2003 [42]	Observational study	Preterm neonates	15	NIRO 300	Day 1–3 after birth	Before and after NIRS measurements	30 min	Day 1: 57% Day 2: 66.1% Day 3: 76.1%	n.r.	n.r.
von Siebenthal K., 2005 [43]	Observational study	Preterm neonates	28	Critikon Cerebral Oxygenation Monitor 2020	Hours 0–6 after birth	n.r.	n.r.	n.r.	7.26	No
Weiss M., 2005 [44]	Prospective observational	Preterm and term neonates	155	NIRO 300	Day 12 (0–365) after birth	During NIRS measurements	30 min in 1 min intervals	60.5%	7.39	No
Victor S., 2005 [45]	Prospective observational study	Preterm neonates	22	NIRO 500	Day 1–3 after birth	Midway through an EEG recording	Once a day during EEG measurement from day 1 to 3 after birth	n.r.	Day 1: 7.3 Day 2: 7.3 Day 3: 7.3	n.r.
van Alfen-van der Velden A.A.E.M., 2006 [46]	Randomized controlled study	Preterm neonates	29	OXYMON	n.r.	Before and 30 min after completion of HCO_3_ administration	From 10 min before until 45 min after HCO_3_ administration	n.r.	Before: 7.29 and 7.29; After: 7.33 and 7.35	No
Zaramella P., 2006 [47]	Observational study	Preterm neonates	16	NIRO-300	Day 7–33 after birth	35th min before and 27th min after surgical manoeuvres	27th min before and 14th and the 35th min after surgical manoeuvres	35 min before manoeuvres: 61.1%; 14 min after manoeuvres: 56.6%; 27 min after manoeuvres: 55.8%	35 min before manoeuvres: 7.27; 27 min after manoeuvres: 7.35	n.r.
Victor S., 2006 [48]	Prospective observational study	Preterm neonates	40	NIRO 500	Day 1–4 after birth	During NIRS measurements	One hour every day during the first four days after birth	Day 1: FTOE 0.35; Day 2: FTOE 0.29; Day 3: FTOE 0.30; Day 4: FTOE 0.30	LVO: 7.33 RVO: 7.33	n.r.
Zaramella P., 2007 [49]	Case–control study	Preterm and term neonates	22	NIRO-300	Day 1 after birth	Within 1 h after birth	n.r.	Depressed/asphyxiated group: 75.3%; control group: 66.5%; normal 1-year outcome: 74.7%; abnormal 1-year outcome: 80.1%	n.r.	n.r.
Horvath R., 2009 [50]	Retrospective study	Neonates, infants and children	36	INVOS 5100B	Day 10 (1–510) after birth	24 h before, during and 24 h after chest closure	24 h before, during, and 24 h after chest closure	Before chest closure: 62.4%; after chest closure: 56.9%	Before chest closure: 7.41 after chest closure: 7.41	n.r.
Bishay M., 2011 [51]	Prospective observational cohort study	Neonates and infants	8	INVOS	Day 0–314 after birth	Preoperatively, start, during, and end of surgery, 12 and 24 h postoperatively	Preoperatively, start, during, and end of surgery, 12 and 24 h postoperatively	Start: 87%; end: 75%; 12 h post-surgery: 74%; 24 h post-surgery: 73%	Start: 7.19; intraoperatively: 7.05; end: 7.28	n.r.
Gunaydin B., 2011 [52]	Prospective randomized study	Term neonates	90	INVOS 5100	Min 0–5 after birth	After the delivery	1 interval during first 5 min after birth	n.r.	U.A. pH: 7.31, 7.29, and 7.24; U.V. pH: 7.35, 7.35, and 7.29	n.r.
Redlin M., 2011 [53]	Retrospective study	Neonates	23	NIRO 200	Day 2–17 after birth	Pre- and postoperatively, beginning, 15 min intervals during and end of CPB	Continuously before and after surgery and CPB	n.r.	Before surgery: 7.43 and 7.48; start CPB: 7.38 and 7.42; during CPB: 7.39 and 7.40; end of CPB: 7.40 and 7.43; after CPB: 7.41 and 7.44	n.r.
Bravo M.D.C., 2011 [54]	Prospective uncontrolled case series observational study	Neonates and infants	16	NIRO-300	Day 5–42 after birth	Beginning and end of the study	Continuously during 48 h in 20 s intervals	Δ –2.56%	Initial: 7.36; final: 7.42	n.r.
Quarti A., 2011 [55]	Prospective observational study	Neonates, infants, children, and adults	40	INVOS 5100C	Year 8.4 (11 days—60 years)	Before CPB, during cooling, re-warming, weaning, and after CPB	During cardiopulmonary bypass surgery	n.r.	n.r.	n.r.
Amigoni A., 2011 [26]	Prospective observational study	Neonates, infants, and children	16	INVOS 5100C	Month 3.5 (0–66) after birth	Before and after surgical procedure and at start, middle, and end of CPB	Continuously during surgical procedure	Basal 55%; before CPB 42%; CPB start 42.5%; CPB middle 40.5%; CPB before stop 41%; CPB re-warming 46%; after CPB 42.5%; before discharge 50%	Basal: 7.41; CBP start: 7.45; CBP middle: 7.41; CPB before stop: 7.39; after CPB: 7.38	Yes Negative
Quarti A., 2013 [56]	Prospective observational study	Neonates and infants	19	INVOS 5100C	Day 26 (6–120) after birth	Before CPB, at CPB starting, before and during CO_2_ flooding, after stopping CO_2_, and at the end of CPB	Before CPB, at CPB starting, before and during CO_2_ flooding, after stopping CO_2_ and at the end of CPB	Before CPB 54.7%; during CPB 47.7%; before CO_2_ 52.9%; during CO_2_ 63.4%; after CO_2_ 55.8%; after CPB 51.9%	Before CPB: 7.36; during CPB: 7.54; before CO_2_: 7.50; during CO_2_: 7.41; after CO_2_: 7.46; after CPB: 7.38	n.r.
Menke J., 2014 [57]	Prospective observational study	Neonates, infants, and children	10	Critikon Cerebral RedOx Monitor 2020	Year 0–9	5 to 20 min intervals	Continuously before and during CPB surgery	60.0%	7.39	No
Pellicer A., 2013 [58]	Pilot, phase 1 randomized, blinded clinical trail	Neonates and infants	20	NIRO 300	Day 6–34 after birth	Before surgery, 6 h intervals during 24 h, 48, and 96 h	Immediately after surgery and continuously during the first day, for 4 h at 48 and 96 h post-surgery	n.r.	n.r.	n.r.
Conforti A., 2014 [59]	Prospective observational study	Preterm and term neonates	13	INVOS 5100C	n.r.	Preoperatively, interoperatively, end of surgery, 24 and 48 h postoperatively	Continuously 12 h before to 48 h after surgery	n.r.	n.r.	n.r.
Mintzer J.P., 2014 [60]	Prospective observational non-interventional study	Preterm neonates	23	INVOS 5100C	Day 3 (0–7) after birth	Before and after RBC transfusion	Continuously during the 7 days	RBC transfused vs. non-transfused Pre-RBC 69% vs. 79% Post-RBC 76% vs. 79% Post-RBC (24 h) 68% vs. 75%	RBS transfused group: 7.28; non-Transfused group: 7.33	n.r.
Kim J.W., 2014 [61]	Retrospective study of prospective data	Neonates, infants, and children	73	INVOS 5100B	Month 3 (0.1–72)	After separation from CPB	Continuously after induction of anesthesia in a 5 min period	57%	7.35	n.r.
Gupta P., 2014 [62]	Retrospective observational study	Neonates	15	n.r.	Day 19 (12–22) after birth	Before extubation	6 h before and 6 h after extubation	Extubation failure: 56.0% and 57.0%; extubation success: 61.0% and 63.0%	Extubation failure: 7.4 and 7.4; extubation success: 7.4 and 7.4	n.r.
Mintzer J.P., 2015 [63]	Prospective observational cohort study	Preterm neonates	12	INVOS 5100C	Day 1 to 7 after birth	During NIRS measurements	Continuously 1 h prior and 2 h immediately following procedure	74%	Before: 7.23; after: 7.31	No
Mebius M.J., 2016 [64]	Retrospective study	Preterm and term neonates	56	INVOS 4100c and 5100c	Day 0–3 after birth	Daily	Continuously within the first 72 h after birth	Day 1. 58.5% Day 2. 62.5% Day 3. 61.5%	7.29 and 7.31	No
Tytgat S.H.A.J., 2016 [65]	Single-center prospective observational study	Preterm and term neonates	15	INVOS 4100-5100	Days 2 (1–7) after birth	Baseline, after anesthesia induction, after CO_2_-insufflation, before CO_2_ exsufflation, and postoperatively 6, 12, and 24 h	Continuously at baseline, after anesthesia induction, 30 min after CO_2_ insufflation, 30 min before exsufflation, postoperatively 6, 12 and 24 h	After anesthesia induction: 77%; before CO_2_ insufflation: 73%	Baseline: 7.33; after CO_2_-insufflation: 7.25	n.r.
Torres S., 2016 [66]	Prospective pilot study	Neonates, infants, and young children	31	INVOS 5100C	Day 11–2433 after birth	T0: During calibration T1: 5 min after aortic cross-clamp T2: 5 min after test start T3: At the end of the 20 min test T4: After clamp removal	Every 5 min during surgery on left and right hemisphere	Left Right T0: 55.3–55.2% T1: 55.8–55.1% T2: 53.9–52.9% T3: 55.3–54.2% T4: 55.5–54.8%	T0: 7.42 T1: 7.45 T2: 7.44 T3: 7.45 T4: 7.42	n.r.
Dix L.M.L., 2017 [67]	Retrospective observational study	Preterm neonates	38	INVOS 5100C	Day 0–3 after birth	n.r.	Before, during and after fluctuation of CO_2_	Before: 66.0% and 69.6%; during: 71.1% and 61.9%; after: 66.8% and 68.4%	n.r.	n.r.
Hunter C.L., 2017 [68]	Prospective observational study	Preterm neonates	22	NONIN SenSmart X-100 oximetry system	Day 6.2 (1–36) after birth	Single time point during NIRS measurements	10 min before and after blood sample	Between 70% and 80%	n.r.	No
Nissen M., 2017 [69]	Prospective observational study	Preterm and term neonates and infants	12	INVOS 5100C	Day 43 (20–74) after birth	During NIRS measurements, once before restoration, and before and after surgery	Before restoration of metabolic alkalosis, 3 h before, 16 and 24 h after surgery in 30 min intervals	Before restoration 72.74%; before surgery 77.89%; after surgery 80.79%	n.r.	Yes negative
Neunhoeffer F., 2017 [70]	Prospective observational study	Neonates and infants	15	O2C device	Day 5 (1–150) and day 37 (1–68) after birth	Before operation, half-hourly during operation, and after surgery	Continuously before, during and after surgery	Before: 61.85% vs. 65.02%; during: 66.75% vs. 67.62%; after: 66.75% vs. 69.87%	Before: 7.38 vs. 7.39; during: 7.3 vs. 7.38; after: 7.32 vs. 7.34	n.r.
Weeke L.C., 2017 [71]	Observational retrospective cohort study	Preterm and term neonates	25	INVOS 4100-5100	Hour 120 (46.5–441.4) and hour 20.7 (7.2–131) after birth	4 h intervals	Continuously 10 min before, during and/or after hypercapnia	Before: 66.54%; during: 68.36%; after: 65.91%	Before 7.26; during 7.02; after 7.27	n.r.
Katheria A.C., 2017 [72]	Retrospective study	Preterm neonates	36	FORE-SIGHT	Day 1 after birth	Before and 1 h within HCO_3_ administration	Continuously in 10 min periods before, during and after HCO_3_ administration	n.r.	Before: 7.23; after: 7.28	Yes Positive
Mukai M., 2017 [73]	n.r.	Term neonates	35	KN-15, ASTEM	From the second stage of labor to 5 min after birth	During NIRS measurement	Continuously during second stage of labor, crowning, immediately after birth, after the first cry, 1,3, and 5 minutes after the delivery	Second stage of labor: 50.3%; crowning: 32.7%; immediately after birth: 30.0%; after the first cry: 31.6%; 1 min: 50.6%; 3 min: 54.4%; 5 min: 56.8%	7.297	n.r.
Janaillac M., 2018 [74]	Prospective observational study	Preterm neonates	20	INVOS 5100	Day 0–3 after birth	During NIRS measurements every 6 to 8 h	Continuously for 72 h in 30 min intervals	6 h: 69% 24 h: 76% 48 h: 71% 72 h: 68%	6 h: 7.29 24 h: 7.28 48 h: 7.25 72 h: 7.27	No
Mebius M.J., 2018 [75]	Prospective observational study	Term neonates	6	n.r.	Day 0–3 after birth	n.r.	n.r.	Day 1: 77.5% and 0.19 Day 2: 82% and 0.12 Day 3: 78% and 0.16	n.r.	No
Polavarapu S.R., 2018 [76]	Prospective cohort study	Preterm neonates	47	INVOS 5100C	Day 1 to 4 after birth	Cord blood analysis at time of delivery	Continuously during the first 96 h after birth	n.r.	U.A. pH: 7.24; U.V. pH: 7.30	n.r.
Beausoleil T.P., 2018 [77]	Prospective observational study	Preterm neonates	19	INVOS 5100	Day 0–3 after birth	During NIRS measurements every 6 to 8 h	Continuously during the first 72 h after birth	n.r.	PH-IVH: 7.24 Healthy controls: 7.28	n.r.
Costerus S., 2019 [78]	Prospective observational pilot study	Term neonates	10	INVOS 5100C	Day 1.3–4.5 after birth	Baseline, every 30 min during surgery of CDH and EA	Baseline, every 30 min during insufflation	CDH baseline 82% EA baseline 91%	n.r.	n.r.
Leroy L., 2021 [79]	Prospective observational study	Term neonates	20	NIRO-200NX	Minute 2 to 10 after birth	Immediately after birth	3 min to 10 min after birth	n.r.	7.28	Yes negative
Loomba R.S., 2022 [80]	Retrospective single-center study	n.r.	23	FORE-SIGHT	Month 15.4 ±30.8	Baseline, 1 hour after HCO_3_ administration, and 2 hours after HCO_3_ administration	Baseline, 1 h after HCO_3_ administration, 2 h after HCO_3_ administration	Baseline: 64% 1 h after HCO_3_ administration: 65%; 2 h after HCO_3_ administration: 65%	Baseline: 7.24; 1 h after HCO_3_ administration: 7.31; 2 h after HCO_3_ administration: 7.30	No
Knieling F., 2022 [81]	Prospective single-center cross-sectional diagnostic study	Term neonates	12	n.r.	Day 6.9 (2–16) after birth	Before (T1) and after (T5) surgery, during high flow of the CPB at 37 °C (T2), and at 25–28 °C (T3), during low flow of the CPB at 2–28 °C (T4)	Before (T1) and after (T5) surgery, during high flow of the CPB at 37 °C (T2), & at 25–28 °C (T3), during low flow of the CPB at 25–28 °C (T4)	T1: 44% T2: 53% T3: 67% T4: 62% T5: 76%	T1: 7.4 T2: 7.4 T3: 7.3 T4: 7.2 T5: 7.4	n.r.
Savorgnan F., 2023 [82]	Single-center, retrospective analysis	Term neonates	134	n.r.	Day 7 (4–10) after birth	Baseline and within 6 h before extubation	Baseline, 10 min after extubation & 120–180 min post-extubation	Baseline: 57.9%; 10 min after extubation: −1.7% × min; 120–180 min post-extubation: −0.4% × min	Baseline: 7.38; within 6 h before extubation: 7.40	n.r.
Mattersberger C., 2023 [34]	Prospective observational study	Preterm and term neonates	157	INVOS 5100	Min 15 after birth	Between 10 to 20 min after birth	Continuously at 15th minute after birth	Preterm neonates: 82%; term neonates: 83%; preterm neonates: 0.13; term neonates: 0.14	Preterm neonates: 7.267; term neonates: 7.293	No in term neonates Yes in preterm neonates; crSO_2_: positive; FTOE: negative
Kazanasmaz H., 2023 [83]	Prospective observational study	Term neonates	84	MASIMO O3	Hour < 6 after birth	Immediately after birth	Continuously for 10 min before starting therapeutic hypothermia	PA group: 67% and 67%; control group: 80% and 79%	6.93	Yes Positive
Cheng K. 2024 [84]	Randomized control study	Preterm neonates	98	n.r.	Day 4 to 9 after birth	Day 0–5	within the first 72 h, 96 h and 120 h	71.15%	7.25	n.r.
Dusleag M. 2024 [85]	Prospective observational study	Preterm and term neonates	77	INVOS 5100	During the first 15 min after birth	Immediately after birth	during the first 15 min after birth	Preterm neonates: 44%; term neonates: 62.2%	Preterm neonates: 7.32; term neonates: 7.32	No

n.r. = Not reported; crSO_2_ = cerebral regional oxygen saturation; FTOE = fractional tissue oxygen extraction; HCO_3_ = bicarbonate; CPB = cardiopulmonary bypass; NIRS = near-infrared spectroscopy; CO_2_ = carbon dioxide; U.A. = umbilical artery; U.V. = umbilical venous; CDH = congenital diaphragmatic hernia; EA = esophageal atresia; PH-IVH = pulmonary and intraventricular hemorrhage; RBC = red blood cell; min = minutes; RVO = right ventricular output; TOI = tissue oxygenation index; LVO = left ventricular output; PA = perinatal asphyxia; Δ = delta.

**Table 2 children-12-01549-t002:** Base excess/base deficit and cerebral oxygenation in the neonatal period.

First Author, Years	Study Design	Neonates	n	Device	NIRS Measurement, Time Point	Blood Sample, Time Point	NIRS Measurement, Duration	TOI or crSO_2_ or FTOE	Base Excess or Base Deficit, Mean Value	Association, Correlation
Aldrich C.J., 1994 [38]	Prospective observational study	Fetus at delivery	33	n.r.	During the delivery	Immediately after birth	10 min period 30 min before birth	n.r	n.r.	Yes Negative
Ramamoorthy C., 2002 [40]	Randomized crossover trial	Preterm and term neonates	15	NIM	Day 3 (2–14) after birth	At the end of base; 17% fractional inspired O_2_; second base; and 3% fractional inspired CO_2_	10–20 min period at base; during 17% fractional inspired O_2_; second base; and during 3% fractional inspired CO_2_	At base: 53%l 17% fractional inspired O_2_: 53%; second base: 56%; 3% fractional inspired CO_2_: 68%	At base: 0.3; 17% fractional inspired O_2_: 0.1; second base: 0.6; 3% fractional inspired CO_2_: 1.3	n.r.
Andropoulos D.B., 2003 [41]	Prospective observational study	n.r.		INVOS 5100	Day 13 (2–128) after birth	Every 10 to 20 min during bypass	At baseline full cardiopulmonary bypass flow; during low-flow cerebral perfusion; and after repair full flow	At baseline: 87%; during low-flow cerebral perfusion: 88%; after repair full flow: 86%	At baseline: +1.0; during low-flow cerebral perfusion: +0.1; after full-flow repair: −1.5	n.r.
Weiss M., 2005 [44]	Prospective observational study	Preterm and term neonates	155	NIRO 300	Day 12 (0–365) after birth	During NIRS measurements	30 min in 1 min intervals	60.5%	0.8 mmol/L	No
Victor S., 2005 [45]	Prospective observational study	Preterm neonates	22	NIRO 500	Day 1–3 after birth	Midway through an EEG recording	Once a day during EEG measurement from day 1 to 3 after birth	n.r.	−1.9	n.r.
van Alfen-van der Velden A.A.E.M., 2006 [46]	Randomized controlled study	Preterm neonates	29	OXYMON	n.r.	Before and 30 min after completion of HCO_3_ administration	From 10 min before until 45 min after HCO_3_ administration	n.r.	Before: −7.6 l/l and −7.5 l/l; after: −4.3 l/l and −4.1 l/l	No
Victor S., 2006 [48]	Prospective observational study	Preterm neonates	40	NIRO 500	Day 1–4 after birth	Once during NIRS measurements	1 h every day during the first four days after birth	Day 1. FTOE: 0.35 Day 2. FTOE: 0.29 Day 3. FTOE: 0.30 Day 4. FTOE: 0.30	2.0 (9.6 to −1.3)	n.r.
Zaramella P., 2007 [49]	Case–control study	Preterm and term neonates	22	NIRO-300	Day 1 after birth	Within 1 h after birth	n.r.	Depressed/asphyxiated group: 75.3%; control group: 66.5%; neonates with normal one-year outcome: 74.7%; neonates with abnormal one-year outcome: 80.1%	n.r.	n.r.
Horvath R., 2009 [50]	Retrospective study	Neonates, infants and children	36	INVOS 5100B	Day 10 (1–510) after birth	24 h before chest closure, during chest closure, and 24 h after chest closure	24 h before chest closure; during chest closure; and 24 h after chest closure	Before chest closure: 62.4%; after chest closure: 56.9%	Before chest closure: 2.1; after chest closure: 1.5	n.r.
Gunaydin B., 2011 [52]	Prospective randomized study	Term neonates	90	INVOS 5100	Min 0–5 after birth	After the delivery	1 interval during first 5 min after birth	n.r.	U.A. BE: −2.52, −2.4, and −4.3 mmol/L; U.V. BE: −3.27, −3.18, and −4.8 mmol/L	n.r.
Redlin M., 2011 [53]	Retrospective study	Neonates	23	NIRO 200	Day 2–17 after birth	Pre- and postoperatively At the beginning; 15 min intervals during and at the end of CPB	Continuously before and after surgery and CPB	n.r.	Before surgery: −0.5 and 0.5 mmol/L; start CPB: −4.4 and −2.0 mmol/L; during CPB: −4.1 and −1.7 mmol/L; end of CPB: −3.9 and −0.2 mmol/L; after CPB: −3.2 and −0.6 mmol/L	n.r.
Bravo M.D.C., 2011 [54]	Prospective, uncontrolled, case series observational study	Neonates and infants	16	NIRO-300	Day 5–42 after birth	Beginning and end of the study	Continuously during 48 h in 20 s intervals	Δ –2.56%	Initial: 0.29; final: 2.6	n.r.
Menke J., 2014 [57]	Prospective observational study	Neonates and children	10	Critikon Cerebral RedOx Monitor 2020	Years 0–9	5–20 min intervals	Continuously before and during CPB surgery	60.0%	n.r.	n.r.
Mintzer J.P., 2014 [60]	Prospective observational non-interventional study	Preterm neonates	23	INVOS 5100C	Day 3 (0–7) after birth	60 min before, during, and 120 min after RBC transfusion	Continuously during the seven days	RBC transfused vs. non-transfused Pre-RBC 69% vs. 79% Post-RBC 76% vs. 79% Post-RBC (24 h) 68% vs. 75%	RBS transfused group: 4.8 mmol/L; non-transfused group: 4.9 mmol/L	n.r.
Gupta P., 2014 [62]	Retrospective observational study	Neonates	15	n.r.	Day 19 (12–22) after birth	Before extubation	6 h before and 6 h after extubation	Extubation failure: 56.0% and 57.0%; extubation success: 61.0% and 63.0%	Extubation failure: 3.7 and 2.1; extubation success: 3.1 and 1.4	n.r.
Mintzer J.P., 2015 [63]	Prospective observational cohort study	Preterm neonates	12	INVOS 5100C	Day 3 (2–5) after birth	During NIRS measurements	Continuously 1 h prior and 2 h immediately following procedure	74%	Before: 7.6; after 3.4	No
Torres S., 2016 [66]	Prospective pilot study	Neonates, infants, and young children	31	INVOS 5100C	Day 11–2433 after birth	T0: during calibration; T1: 5 min after aortic cross-clamp; T2: 5 min after test start; T3: at the end of the 20 min test; T4: after clamp removal	Every 5 min during surgery on left and right hemispheres	Left Right T0: 55.3–55.2% T1: 55.8–55.1% T2: 53.9–52.9% T3: 55.3–54.2% T4: 55.5–54.8%	T0: −1.1 mmol/L T1: 0.1 mmol/L T2: 0.9 mmol/L T3: 0.5 mmol/L T4: 0.3 mmol/L	n.r.
Hunter C.L., 2017 [68]	Prospective observational study	Preterm neonates	22	NONIN SenSmart X-100 oximetry system	Day 6.2 (1–36) after birth	Single time point during NIRS measurements	10 min before and after blood sampling	Between 70% and 80%	n.r.	No
Nissen M., 2017 [69]	Prospective observational study	Preterm and term neonates and infants	12	INVOS 5100C	Days 43 (20–74) after birth	During NIRS measurements, once before restoration, and before and after surgery	Before restoration of metabolic alkalosis; 3 h before; 16 and 24 h after surgery in 30 min intervals	Before restoration: 72.74%; before surgery: 77.89%; after surgery: 80.79%	n.r.	Yes Negative
Weeke L.C., 2017 [71]	Observational retrospective cohort study	Preterm and term neonates	25	INVOS 4100-5100	Hour 120 (46.5–441.4) and hour 20.7 (7.2–131)	4 h intervals	Continuously 10 min before; during and/or after hypercapnia	Before: 66.54%; during: 68.36%; after: 65.91%;	Before: −4.39 mmol/L; during: −5.39 mmol/L; after: −4.22 mmol/L	n.r
Katheria A.C., 2017 [72]	Retrospective study	Preterm neonates	36	FORE-SIGHT	Day 1 after birth	Before and 1 h within HCO_3_ administration	Continuously in 10 min periods before, during, and after HCO_3_ administration	n.r.	Before: −8.9; After: −6.8	Yes Negative
Costerus S., 2019 [78]	Prospective observational pilot study	Term neonates	10	INVOS 5100C	Day 1.3–4.5 after birth	Baseline, every 30 min during surgery of CDH and EA	Baseline and every 30 min during insufflation	CDH baseline: 82%; EA baseline: 91%	n.r.	n.r.
Leroy L., 2021 [79]	Prospective observational study	Term neonates	20	NIRO-200NX	Minute 1.75 after birth	Immediately after birth	From minute 3 to 10 after birth	n.r.	−2.3 mmol/L	Yes Negative
Mattersberger C., 2023 [34]	Prospective observational study	Preterm and term neonates	157	INVOS 5100	During and immediately after the delivery	Between 10 and 20 min after birth	Continuously at 15th minute after birth	Preterm neonates 82% and term neonates 83%; preterm neonates 0.13 and term neonates 0.14	Preterm neonates—2.3 mmol/L Term; neonates—0.9 mmol/L	No in term neonates; Yes in preterm neonates crSO_2_; positive FTOE; negative
Kazanasmaz H., 2023 [83]	Prospective observational study	Term neonates	84	MASIMO O_3_	Hour < 6 after birth	Immediately after birth	Continuously for 10 min before starting therapeutic hypothermia	PA group: 67% and 67% Control group: 80% and 79%	17.8 mmol/L	No
Dusleag M. 2024 [85]	Prospective observational study	Preterm and term neonates	77	INVOS 5100	During the first 15 min after birth	Immediately after birth	during the first 15 min after birth	Preterm neonates: 44%; term neonates; 62.2%	Preterm neonates: 0.7 mmol/L; term neonates: 1.37 mmol/L	No

n.r. = Not reported; crSO_2_ = cerebral regional oxygen saturation; FTOE = fractional tissue oxygen extraction; HCO_3_ = bicarbonate; CPB = cardiopulmonary bypass; NIRS = near-infrared spectroscopy; CO_2_ = carbon dioxide; U.A. = umbilical artery; U.V. = umbilical venous; CDH = congenital diaphragmatic hernia; EA = esophageal atresia; RBC = red blood cell; min = minutes; TOI = tissue oxygenation index; Δ = delta.

**Table 3 children-12-01549-t003:** Bicarbonate and cerebral oxygenation in the neonatal period.

First Author, Years	Study Design	Neonates	n	Device	NIRS Measurement, Time Point	Blood Sample, Time Point	NIRS Measurement, Duration	TOI or crSO_2_ or FTOE	HCO_3_, Mean Value	Association, Correlation
Naulaers G., 2002 [39]	Observational study	Preterm neonates	15	NIRO 300	Day 1–3 after birth	Before and after NIRS measurements	30 min	Day 1. 57% Day 2. 66.1% Day 3. 76.1%	n.r.	n.r.
van Alfen-van der Velden A.A.E.M., 2006 [46]	Randomized controlled study	Preterm neonates	29	OXYMON	n.r.	Before and 30 min after completion of HCO_3_ administration	From 10 min before until 45 min after HCO_3_ administration	n.r.	Before: 18.4 mmol/L and 18.3 mmol/L; after: 21.1 mmol/L and 21.0 mmol/L	No
Bravo M.D.C., 2011 [54]	Prospective uncontrolled case series observational study	Neonates and infants	16	NIRO-300	Day 5–42 after birth	Beginning and end of the study	Continuously during 48 h in 20 s intervals	Δ –2.56%	Initial: 27.2 mmol/L; final: 27.2 mmol/L	n.r.
Mintzer J.P., 2015 [63]	Prospective observational cohort study	Preterm and term neonates	12	INVOS 5100C	Day 3 (2–5) after birth	During NIRS measurements	Continuously 1 h prior and 2 h immediately following the procedure	74%	4.5 mL/kg −1	No
Dix L.M.L., 2017 [67]	Retrospective observational study	Preterm neonates	38	INVOS 5100C	Day 0–3 after birth	n.r.	Before, during and after fluctuation of CO_2_	Before: 66.0% and 69.6%; during: 71.1% and 61.9%; after: 66.8% and 68.4%	n.r.	n.r.
Hunter C.L., 2017 [68]	Prospective observational study	Preterm neonates	22	NONIN SenSmart X-100 oximetry system	Day 6.2 (1–36) after birth	Single time point during NIRS measurements	10 min before and after blood sample	Between 70% and 80%	n.r.	No
Nissen M., 2017 [69]	Prospective observational study	Preterm and term neonates and infants	12	INVOS 5100C	Day 43 (20–74) after birth	During NIRS measurements, once before restoration, and before and after surgery	Before restoration of metabolic alkalosis, 3 h before, and 16 and 24 h after surgery in 30 min intervals	Before restoration: 72.74%; before surgery: 77.89%; after surgery: 80.79%	n.r.	Yes Negative
Weeke L.C., 2017 [71]	Observational retrospective cohort study	Preterm and term neonates	25	INVOS 4100-5100	Hour 120 (46.5–441.4) and hour 20.7 (7.2–131)	4 h intervals	Continuously 10 min before and during and/or after hypercapnia	Before: 66.54%; during: 68.36%; after: 65.91%	Before: 22.64 mmol/L; during: 25.14 mmol/L; after: 22.61 mmol/L	n.r
Katheria A.C., 2017 [72]	Retrospective study	Preterm neonates	36	FORE-SIGHT	Day 1 after birth	Before and 1 h within HCO_3_ administration	Continuously in 10 min periods before, during, and after HCO_3_ administration	n.r.	n.r.	Yes Positive
Loomba R.S., 2022 [80]	Retrospective single-center study	n.r.	23	FORE-SIGHT	Months 15.4 ± 30.8	Baseline, 1 h after HCO_3_ administration, and 2 h after HCO_3_ administration	Baseline, 1 h after HCO_3_ administration, and 2 h after HCO_3_ administration	Baseline: 64%; 1 h after HCO_3_ administration: 65% 2 h after HCO_3_ administration: 65%	Baseline: 18 mEq/L; 1 h after HCO_3_ administration: 21 mEq/L; 2 h after HCO_3_ administration: 20 mEq/L	No
Savorgnan F., 2023 [82]	Single-center, retrospective analysis	Term neonates	134	n.r.	Day 7 (4–10) after birth	Baseline and within 6 h before extubation	At baseline, 10 min after extubation, and 120–180 min post-extubation	At baseline: 57.9%; 10 min after extubation: −1.7% × min; 120–180 min post-extubation: −0.4% × min	At baseline: 27 mEq/L; within 6 h before extubation: 27.0 mEq/L	n.r.
Mattersberger C., 2023 [34]	Prospective observational study	Preterm and term neonates	157	INVOS 5100	During and immediately after the delivery	Between 10 to 20 min after birth	Continuously at 15th minute after birth	Preterm neonates 82% and term neonates 83%; preterm neonates 0.13 and term neonates 0.14	Preterm neonates: 21.0 mmol/L; term neonates: 21.6 mmol/L	No in preterm neonates; yes in term neonates; FTOE positive
Dusleag M. 2024 [85]	Prospective observational study	Preterm and term neonates	77	INVOS 5100	During the first 15 min after birth	Immediately after birth	During the first 15 min after birth	Preterm neonates: 44%; term neonates: 62.2%	Preterm neonates: 22.9 mmol/L; term neonates: 23.0 mmol/L	No

n.r. = Not reported; crSO_2_ = cerebral regional oxygen saturation; FTOE = fractional tissue oxygen extraction; HCO_3_ = bicarbonate; NIRS = near-infrared spectroscopy; CO_2_ = carbon dioxide; min = minutes; TOI = tissue oxygenation index; Δ = delta.

**Table 4 children-12-01549-t004:** Quality of included studies as assessed using the Risk of Bias in Non-Randomized Studies of Exposure (ROBINS-E) tool.

Author	Bias Due to Confounding	Bias Arising from Measurement of the Exposure	Bias in Selection of Participants into the Study	Bias Due to Post-Exposure Interventions	Bias Due to Missing Data	Bias Arising from Measurement of the Outcome	Bias in Selection of the Reported Result	Overall Risk of Bias Rating
Aldrich C.J., 1994 [38]	high risk of bias	low risk	low risk	low risk	high risk of bias	low risk	some concerns	very high risk of bias
von Siebenthal K., 2005 [43]	low risk	low risk	some concerns	some concerns	some concerns	low risk	some concerns	some concerns
Weiss M., 2005 [44]	some concerns	low risk	low risk	some concerns	low risk	low risk	some concerns	some concerns
van Alfen-van der Velden A.A.E.M., 2006 [46]	low risk	low risk	low risk	some concerns	low risk	low risk	some concerns	some concerns
Amigoni A., 2011 [26]	high risk of bias	low risk	low risk	some concerns	some concerns	low risk	some concerns	high risk of bias
Menke J., 2014 [57]	high risk of bias	low risk	low risk	some concerns	low risk	low risk	some concerns	high risk of bias
Mintzer J.P., 2015 [63]	high risk of bias	low risk	low risk	some concerns	high risk of bias	low risk	some concerns	very high risk of bias
Mebius M.J., 2016 [64]	high risk of bias	low risk	high risk of bias	low risk	high risk of bias	low risk	some concerns	very high risk of bias
Hunter C.L., 2017 [68]	low risk	low risk	low risk	some concerns	low risk	low risk	some concerns	some concerns
Nissen M., 2017 [69]	high risk of bias	low risk	low risk	some concerns	high risk of bias	low risk	some concerns	very high risk of bias
Katheria A.C., 2017 [72]	high risk of bias	low risk	low risk	some concerns	low risk	low risk	some concerns	high risk of bias
Janaillac M., 2018 [74]	low risk	low risk	some concerns	some concerns	high risk of bias	low risk	some concerns	high risk of bias
Mebius M.J., 2018 [75]	some concerns	low risk	some concerns	some concerns	high risk of bias	low risk	some concerns	high risk of bias
Leroy L., 2021 [79]	low risk	low risk	low risk	low risk	low risk	low risk	some concerns	some concerns
Loomba R.S., 2022 [80]	high risk of bias	low risk	low risk	some concerns	high risk of bias	low risk	some concerns	very high risk of bias
Mattersberger C., 2023 [34]	low risk	low risk	low risk	low risk	high risk of bias	low risk	some concerns	high risk of bias
Kazanasmaz H., 2023 [83]	some concerns	low risk	some concerns	high risk of bias	low risk	low risk	some concerns	high risk of bias
Dusleag M. 2024 [85]	low risk	low risk	low risk	low risk	high risk of bias	high risk of bias	some concerns	very high risk of bias

**Table 5 children-12-01549-t005:** Overview of ROBINS-E (risk-of-bias assessment) and correlations between parameters of the acid–base status and cerebral oxygenation in neonates during the neonatal period.

	ROBINS-E	Correlation Analysis					
		crSO_2_ or FTOE					
Author	Overall Risk-of-Bias Rating	pH		BE or BD		HCO_3_	
Aldrich C.J., 1994 [38]	very high risk of bias	yes	+	yes	-		
von Siebenthal K., 2005 [43]	some concerns	no					
Weiss M., 2005 [44]	some concerns	no		no			
van Alfen-van der Velden A.A.E.M., 2006 [46]	some concerns	no		no		no	
Amigoni A., 2011 [26]	high risk of bias	yes	-				
Menke J., 2014 [57]	high risk of bias	no					
Mintzer J.P., 2015 [63]	very high risk of bias	no		no		no	
Mebius M.J., 2016 [64]	very high risk of bias	no					
Hunter C.L., 2017 [68]	some concerns	no		no		no	
Nissen M., 2017 [69]	very high risk of bias	yes	-	yes	-	yes	-
Katheria A.C., 2017 [72]	high risk of bias	yes	+	yes	-	yes	+
Janaillac M., 2018 [74]	high risk of bias	no					
Mebius M.J., 2018 [75]	high risk of bias	no					
Leroy L., 2021 [79]	some concerns	yes	-	yes	-		
Loomba R.S., 2022 [80]	very high risk of bias	no				no	
Mattersberger C., 2023 [34]	high risk of bias	yes	+	yes	+	yes	
Kazanasmaz H., 2023 [83]	high risk of bias	yes	+	no			
Dusleag M. 2024 [85]	very high risk of bias	no		no		no	

BD = Base deficit; BE = base excess; HCO_3_ = bicarbonate; crSO_2_ = cerebral regional oxygen saturation; FTOE = fractional tissue oxygen extraction.

## Data Availability

No new data were created.

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
