# Peer review of "Acid–Base Status and Cerebral Oxygenation in Neonates: A Systematic Qualitative Review of the Literature"

_children, 2025, doi:10.3390/children12111549_

Round 1
Reviewer 1 Report
Comments and Suggestions for Authors
This systematic review by Mattersberger and colleagues summarised the state of art regarding the relationship between blood gas acid-base status and cerebral oxygenation monitored by near-infrared spectroscopy.
Abstract:
Well written with the synopsis of the study project as well as the necessity of investigation in this field.
Backgrounds:
The authors are encouraged to include a section which illustrates the theoretically expected reaction of cerebral vascular system to changes in pH and CO2 so that the audience can understand which finding is under/out of expectation. Also the tricky side (i.e. Sat changes caused by changes of other than vascular reaction) of NIRS-based cerebral oximetry needs to be touched either in the Backgrounds or Discussion section.
Methods:
The study design appears sound, however, I leave its assessment to other expert reviewers as I am not particular bright in it.
Results:
Tables summarising the findings of cited studies are very informative. The authors observed controversial findings with both positive and negative correlations between acid base measures and NIRS measures, which were predominantly seen in studies with relatively high risk of bias. It is possible that studies involving more fragile infants, such as preterm newborns, may inevitably lead to high risk of bias due to difficulties in punctual data sampling. Please consider assessing in depth whether the risk of bias is associated with specific characteristics of the study itself (e.g. age, cohort, and others). Otherwise, it is too preliminary to conclude that the risk of bias caused related studies to mislead their type-1 error based positive/negative correlations.
Discussion:
Once again, it is essential that readers understand theoretically expected vascular reactions to changes in pH, PCO2 and HCO3-. Descriptions such as “the cardiovascular system significantly influences cerebral tissue oxygenation in neonates” falls very short from what readers may expect. The authors are encouraged to include the basic knowledge regarding the cerebral vascular autoregulation. This may also help expand the discussion in 4.4.
Also for section 4.7, as previously mentioned, it is too preliminary to conclude that the bias was responsible to false-positive findings. Together with the discussion in sections 4.5 and 4.6, please discuss in depth why such correlations happened to be noted.
Despite these criticisms shortlisted, I consider this article will encourage future studies focusing on the relationship between cerebral circulation/oxygenation and blood gas control, and therefore, is worth being published in Children with substantial revisions.
Author Response
Response to the Reviewers
We would like to thanks the Reviewers for reviewing our manuscript “Acid-base Status and cerebral Oxygenation in Neonates: A Systematic Qualitative Review of Literature” for publication. We have made the following changes according to the Reviewer’s suggestions:
Reviewer 1
Comments and Suggestions for Authors:
This systematic review by Mattersberger and colleagues summarised the state of art regarding the relationship between blood gas acid-base status and cerebral oxygenation monitored by near-infrared spectroscopy.
Response: The reviewer is correct.
Abstract:
Well written with the synopsis of the study project as well as the necessity of investigation in this field.
Backgrounds:
The authors are encouraged to include a section which illustrates the theoretically expected reaction of cerebral vascular system to changes in pH and CO2 so that the audience can understand which finding is under/out of expectation. Also the tricky side (i.e. Sat changes caused by changes of other than vascular reaction) of NIRS-based cerebral oximetry needs to be touched either in the Backgrounds or Discussion section.
Response: We agree with the reviewer and have included the following passage in the discussion section.
4.7. Proposed explanatory model (Discussion, page 21-22, line 309-334)
… Acidosis can reduce the contractility of cardiomyocytes…
Methods:
The study design appears sound, however, I leave its assessment to other expert reviewers as I am not particular bright in it.
Results:
Tables summarising the findings of cited studies are very informative. The authors observed controversial findings with both positive and negative correlations between acid base measures and NIRS measures, which were predominantly seen in studies with relatively high risk of bias. It is possible that studies involving more fragile infants, such as preterm newborns, may inevitably lead to high risk of bias due to difficulties in punctual data sampling. Please consider assessing in depth whether the risk of bias is associated with specific characteristics of the study itself (e.g. age, cohort, and others). Otherwise, it is too preliminary to conclude that the risk of bias caused related studies to mislead their type-1 error based positive/negative correlations.
Response: The reviewer is correct. We have already noted that the majority of studies present a high to very high risk of bias, and that this represents only one possible factor contributing to a potential association.
…These methodological weaknesses may contribute to the incongruent results regarding potential associations (Discussion, page 22, line 344-345)
… Variability in findings can be attributed to heterogeneity in study populations especially concerning gestational age, timing of measurements, and clinical contexts… (Conclusions, page 22, line 348-350)
Discussion:
Once again, it is essential that readers understand theoretically expected vascular reactions to changes in pH, PCO2 and HCO3-. Descriptions such as “the cardiovascular system significantly influences cerebral tissue oxygenation in neonates” falls very short from what readers may expect. The authors are encouraged to include the basic knowledge regarding the cerebral vascular autoregulation. This may also help expand the discussion in 4.4.
Response: We agree with the reviewer and have included the following passage in the discussion section.
4.7. Proposed explanatory model (Discussion, page 21-22, line 309-334)
Also for section 4.7, as previously mentioned, it is too preliminary to conclude that the bias was responsible to false-positive findings. Together with the discussion in sections 4.5 and 4.6, please discuss in depth why such correlations happened to be noted.
Response: The reviewer is correct. As mentioned above, this factor has already been highlighted in the “Risk of Bias Assessment” section.
…These methodological weaknesses may contribute to the incongruent results regarding potential associations (Discussion, page 22, line 344-345)
Despite these criticisms shortlisted, I consider this article will encourage future studies focusing on the relationship between cerebral circulation/oxygenation and blood gas control, and therefore, is worth being published in Children with substantial revisions.

Reviewer 2 Report
Comments and Suggestions for Authors
Dear Authors,
I have read an article regarding a systematic review of acid-base status and cerebral oxygenation in neonates. Several points need addressing:
1) The list of abbreviations needs updating. There are a lot of abbreviations usedthat are missing, such as PVL
2) Please include the aim of the study at the end of the introduction
3) The PROSPERO registration number (CRD420250655009) could not be traced back to the website. Please revise the ID number.
4) Please indicate which version of PRISMA the authors are using. The authors are citing the 2009 version, while the 2020 version is out there. Similarly, the PRISMA flow chart belongs to the 2009 version and not the 2020 version.
5) Please include the PRISMA 2020 checklist as the supplementary material
6) What does TOI in table 1 stand for?
Author Response
Reviewer 2
We would like to thanks the Reviewers for reviewing our manuscript “Acid-base Status and cerebral Oxygenation in Neonates: A Systematic Qualitative Review of Literature” for publication. We have made the following changes according to the Reviewer’s suggestions:
Dear Authors,
I have read an article regarding a systematic review of acid-base status and cerebral oxygenation in neonates. Several points need addressing:
1) The list of abbreviations needs updating. There are a lot of abbreviations used that are missing, such as PVL
Response: The reviewer is correct. We have updated the list of abbreviations to include all necessary terms and have organised it in alphabetical order. …The following abbreviations are used in this manuscript: (Abbreviations, page 23, line 368)
2) Please include the aim of the study at the end of the introduction
Response: We included the aim of this review at the end of the introduction. …The aim of this review is to provide an overview of the current literature regarding a potential association between parameters of the acid–base status and cerebral oxygenation in neonates during the neonatal period… (Introduction, page 2, line 87-90)
3) The PROSPERO registration number (CRD420250655009) could not be traced back to the website. Please revise the ID number.
Response: Searching for the registration on the PROSPERO website using the registration number is only possible if the sub-folder “PROSPERO registration number AN” is selected.
4) Please indicate which version of PRISMA the authors are using. The authors are citing the 2009 version, while the 2020 version is out there. Similarly, the PRISMA flow chart belongs to the 2009 version and not the 2020 version.
Response: The reviewer is correct. We have used the updated PRISMA guideline for reporting systematic reviews 2020. We have updated our review according to the latest PRISMA flowchart criteria and made the following changes to the manuscript.
…Studies were identified using the stepwise approach outlined in the Preferred Reporting Items for Systematic Reviews and Meta-Analysis (PRISMA 2020)) Statement.[36]…(Materials and Methods, page 3, line 93-95)
… 36. Page MJ, McKenzie JE, Bossuyt PM, et al. The PRISMA 2020 statement: an updated guideline for reporting systematic reviews. Syst Rev. 2021 Mar 29;10(1):89. (References, page 25, line 443-445)
5) Please include the PRISMA 2020 checklist as the supplementary material
Response: We have uploaded the updated PRISMA checklist.
6) What does TOI in table 1 stand for?
Response: TOI reflects regional tissue oxygenation by estimating the ratio of oxygenated haemoglobin to total haemoglobin measured with NIRS. To enhance clarity, we have implemented the following revisions to the manuscript.
…Cerebral oximetry is a continuous, non-invasive, real-time method using near-infrared spectroscopy (NIRS) to detect cerebral tissue oxygenation [(cerebral regional oxygen saturation (crSO2), tissue oxygenation index (TOI) and (fractional tissue oxygen extraction (FTOE)]… (Introduction, page 2, line 62-65)
… TOI = tissue oxygenation index (Table legend, Table 1., page 11, line 178)
… TOI = tissue oxygenation index (Table legend, Table 2., page 14, line 188)
… TOI = tissue oxygenation index (Table legend, Table 3., page 17, line 198)

Round 2
Reviewer 1 Report
Comments and Suggestions for Authors
the authors are praised for improving clarity of the manuscript.
Reviewer 2 Report
Comments and Suggestions for Authors
The authors have satisfactorily addressed all of my concerns.